# Implementation Phase Safety System for Minimising Construction Project Waste

**Kamal Mahfuth [1], Amara Loulizi [1] , Khalid Al Hallaq [2] and Bassam A. Tayeh [2,*]** 

[1] Laboratory of Materials, Department of Civil Engineering, Optimization, and Energy for Sustainability (LAMOED), Engineering School of Tunis, Tunis El Manar University, B.P. 37 Le Belvédère, 1002 Tunis, Tunisia; Engkamal2015@hotmail.com (K.M.); amlouliz@vt.edu (A.L.)

[2] Department of Civil Engineering, Faculty of Engineering, Islamic University of Gaza, 00972 Gaza, Palestine; khalaq168@gmail.com

\* Correspondence: btayeh@iugaza.edu.ps; Tel.: +970-595-174717

**Abstract:** The construction sector is a key component of a nation's gross domestic product, but its inherent nature results in potentially dangerous conditions that affect the safety of all workers on construction projects (CPs). Therefore, the original idea of the research is to determine the relationship between safety system (SS) during the implementation phase (IPh) of CPs and the minimisation of waste (materials, time and cost). Achieving a lean construction work requires suitable planning, safety considerations and waste resource minimisation throughout the project cycle. This research aims to identify and rank the safety factors during the IPh of a CP, which will have positive effects on minimising waste. Information and data were gathered from the existing literature and the structured interviews and questionnaire survey conducted among 111 randomly selected construction companies. Questionnaire results were evaluated using statistical tools, such as hypothesis testing, analysis of variance and linear regression. This research identified and ranked 24 important safety factors with positive effects on minimising waste in CPs during IPh. The seven most important safety factors that should be considered to minimise material, time and cost wastage are as follows: handling, management, external factors, workers, procurement, site condition and appropriate scaffolding for SS. The best linear model was developed on the basis of the importance index of the identified factors. This model can predict the minimisation of waste (materials, time and cost) in CPs by using SS. Thus, the safety criteria and SS should be used during IPh to minimise waste on the basis of the developed model.

**Keywords:** construction safety; construction waste; implementation phase; safety system

## 1. Introduction

Construction projects (CPs) have been identified as one of the most hazardous industries [1–3]. Injuries lead to the suffering of people, unnecessary compensation costs, time overrun, productivity and efficiency reduction, material wastage and increased rate of employee turnover. The 2015 annual report of the International Labour Organization indicated that the cost of poor safety practices accounts for 4% of the annual global gross domestic product [4]. Nahmens and Ikuma [5] stated that a poor safety practice is a form of waste. Therefore, safety is critical for improving productivity and efficiency in CPs. To complete a CP at the lowest cost, highest quality and shortest time, increased attention and commitment must be provided to a safety system (SS) during the implementation stage, and all construction plans must include safety consideration. Additionally, improving occupational safety in CPs is essential not only because enlightened clients demand excellent safety performance from contractors but also due to the continuous search for further economic benefit and increased

productivity [6–9]. The most important problems related to safety at the workplace are as follows: (1) commitment of high management to SS, (2) awareness and training for safety, (3) safety clothes and equipment, (4) enhanced culture and climate of safety and (5) types and numbers of safety staff [10–12]. Therefore, supervisors should be further attentive to these issues.

Waste can affect the success of CPs in terms of cost, time, productivity, sustainability and environment. Construction waste management (CWM) activities are inherent throughout the entire CP cycle—from initial design to demolition. Construction waste (CW) is classified into physical (materials) and non-physical (time and cost) wastes [13–15]. CW focuses not only on material wastage on site but also on any form of inefficiency in productivity, work quality, handling and storage of materials, activity time and workers' movement [16–20].

This study determined the relationship between commitment to SS during IPh and CW (material, time and cost overruns) in CPs. This objective was achieved by identifying and ranking the most important safety factors (24 factors) during the implementation phase (IPh) that have positive effects on waste minimisation and by building a model on the basis of these factors to minimise CW using SS. This research and proposed model contribute to waste (materials, time and cost) reduction in CPs; thus, they have a positive effect on the environment, economy and occupational health in any country. Although the sample used to identify the safety factors belongs to a developing country, the research procedure described in this study could be used for any country.

## 2. Literature Review

The literature search included standard methods (i.e., database search, including ScienceDirect, Google Scholar and TRIS) and the research team's extensive domestic and international contacts for finding pertinent data and citations on the topic that have not been formally published. Additional literature comprised journals, conference papers and books. The data addressed basic concepts and practices in construction safety (CS) and CWM, along with the safety factors that exert positive effects on minimising CW during IPh. A brief description of the major findings from the literature review is presented as follows.

### 2.1. CS Concepts and Definitions

To manage and reduce risk in CPs, an SS should include policies, strategies and procedures, organisational structures, human resource development programmes, control and communication and other safety considerations for each activity on a site [21,22]. The construction industry is one of the most hazardous sectors due to the nature of the work involved, which results from the integration of materials, tools, the environment and various human factors [3,23]. The accident in CPs has one of the highest rates compared with other industries given the most demanding conditions of physical work [24,25]. Statistics indicate that the construction industry still suffers from safety problems. In the US, the average rate of accidents in CPs is three times that in other jobs. The construction sector employs only 7% of workers but accounts for 21% of injuries [26]. In the UK, the average rate of accidents in CP is five times that in other works [27]. In the Palestinian National Authority, construction sector employment has increased from 7.9% in 1970 to 15.5% in 2015, and 37% of work injuries were in CPs [28]. Safety culture in CP refers to how all members in a worksite safely behave, plan and practice any activity [29–32]. A safety management system (SMS) indicates the methodology and regulations for managing the site without dangers. A suitable SMS must contain six elements: policy, strategy and measurements, responsibility for all parties, staff development programmes for safety, coordination, and evaluation and monitoring [21,22,33].

### 2.2. Performance Factors on CPs during IPh

Safety performance in CP is a complex phenomenon because it is a heterogeneous process involving the knowledge and skills of supervisors, behaviour and culture of workers and workplace environment. Accidents in CPs occur due to various reasons: lack of knowledge, training, supervision

or means for safely performing tasks, error of judgment, carelessness, apathy or downright recklessness. In addition, the short-term and transitory nature of the construction industry, the uncontrolled working environment and the complexity and diversity of the size of organisations affect safety performance within CPs [34]. Several considerations for enhancing CS during IPh have been reported by different researchers. In previous studies [9,34–37], 20 factors were identified: (1) scaffolding, (2) ladder access to heights, (3) mobile scaffolds, (4) workplace access, (5) housekeeping, (6) roof work, (7) personal protective equipment, (8) mobile-elevated work platforms, (9) site safety information documents, (10) plan of action, (11) competency of workers and ongoing training, (12) monitoring system, (13) hazard reporting, (14) accident reporting, (15) incidents/near misses, (16) discipline, (17) Health and Safety Authority(HSA) inspections, (18) communication in the workplace, (19) responsibility for safety in the workplace and (20) cooperation. International laws, regulations and specifications have discussed safety requirements in CPs as labour laws [38], such as Occupational Safety and Health Agreement No. 155 [39], Occupational Health Services No. 161 [40], Promotional Framework for Occupational Safety and Health Agreement No. 187 [41] and Health, Safety, and Environmental Minimum Performance Requirements for Contractors [42]. The unacceptable form of work related to occupational safety and health (OSH) in the construction sector are as follows: (1) For medical care, no medical check-up before starting the current job and no periodical medical examination in the workplace [38,43]. (2) For the official inspection system, no effective inspection on OSH measurement in sites and the workplace; no or minimal formal representation from official inspectors on working conditions and OSH; lack of safety system at the workplace, such as risk assessments; and lack of formal representation on working conditions and OSH [44,45]. (3) Lack of suitable personal protective clothing at the workplace [45]. (4) Use of materials that have negative effects on health and exposure to physical and psychological violence at work (e.g., stress, bullying and verbal and sexual harassment) [42,46]. (5) Poor emergency handling, including first-aid arrangements [47]. (6) OSH information is unavailable for labourers, OSH training is not provided and the worker was deprived of the decision to remove himself/herself from possible danger [48]. (7) Use of hazardous equipment that adversely affects the OSH of workers [49]. (8) No suitable alternative employment for workers who cannot continue working under the same occupational hazardous exposure and substituting preventative OSH measures by providing different forms of compensation [50].

## 2.3. CW Concepts and Definitions

The cost, time and productivity in any CP are directly linked with CWM [20]. Several studies from different countries have discussed the conceptualisation of waste in the lean construction philosophy. This conceptualisation is related to the existence of activities without value, including overtime, unnecessary expense of resources or space, unnecessary worker movement, waiting time and rework [11,13,17,51–57]. CW is often clustered into physical (materials wastage) and non-physical (time and cost) overruns. CW includes rework, poor quality, bad planning for workers and any unacceptable form of work [14–16,58–60]. The main factors that generate CW are design or culture, procurement, handling and operation, as summarised in Figure 1 [18,61].

The poor safety in construction projects is a form of wastage of resources. It is costly in many aspects such as human suffering, workers' compensation, time overrun, and loss in productivity. Figure 2 illustrates this concept.

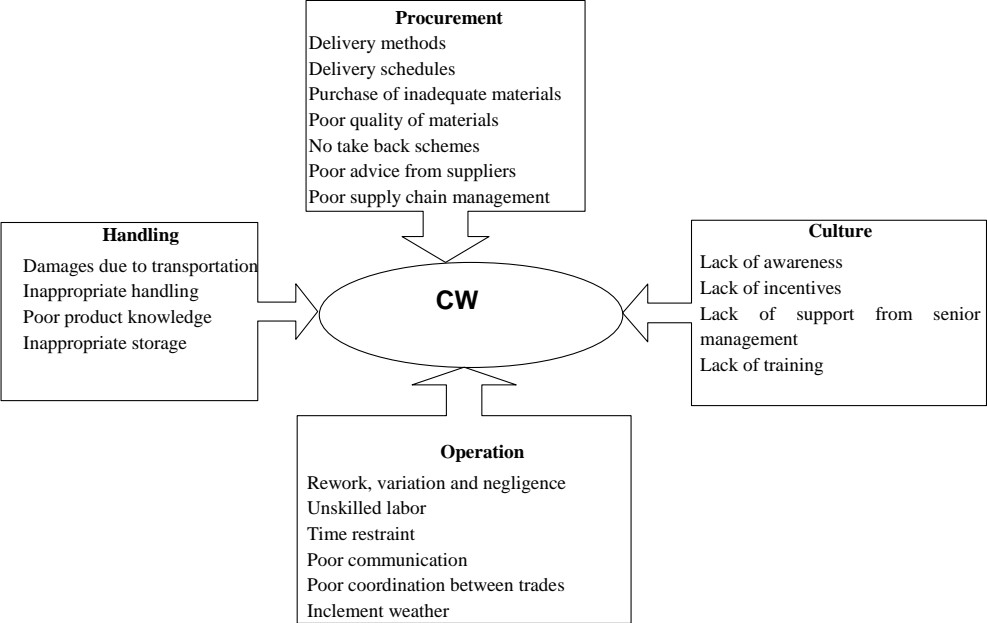

**Figure 1.** Source of Construction waste (CW).

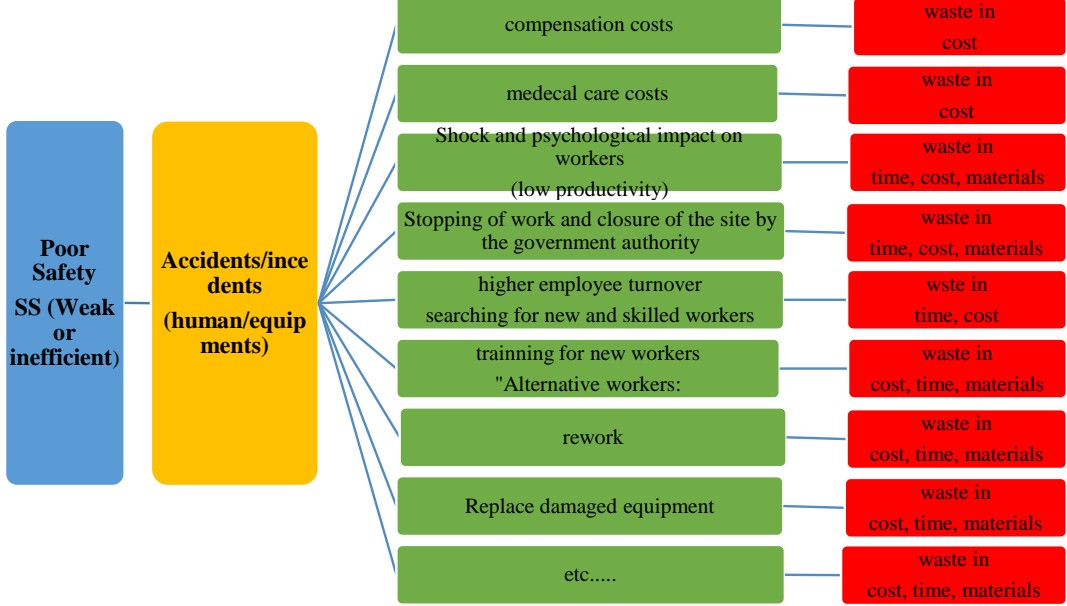

**Figure 2.** Impact of poor safety.

## 3. Research Methodology

As shown in Figure 3, the investigating team relied on the outcome of three information sources to achieve the research objectives. In particular, the findings of the available literature, structured interviews with experts and pilot questionnaires were used to finalise the structure and content of the questionnaire that would be distributed to 111 professionals. The triangulation method, through cross-verification of the three data sources, was adopted to enhance the reliability and validity of the research findings.

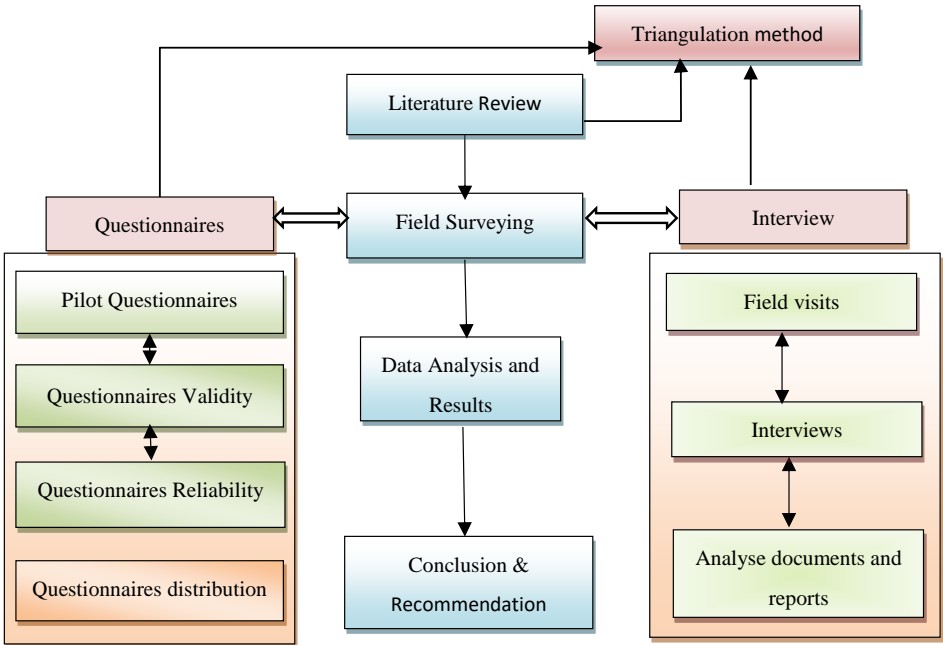

**Figure 3.** Research procedure.

*3.1. Questionnaire's Design*

The questionnaire was chosen to be the main method of collecting data in this research, since the questionnaire is probably the most widely used data collection technique for conducting surveys. Data are collected in a standardized form from samples of population to allow carrying out statistical inferences on the data by computerized programmes. The questionnaire was developed to identify and rank safety factors have positive impacts on minimizing the waste of material, time, and costs during IPh.

The questionnaire was initially designed based on the extensive literature review of previous studies. The first questionnaire draft was designed to be reviewed by a pilot study and, based on the results, the questionnaire framework was modified and developed based on a pilot study, and observations from visiting many projects, experts opinions and structured interviews.

The questionnaire was divided into three main sections, which included general information of respondent (the institution and the participant), safety management practices in the institution and identifying safety factors have positive impacts on minimizing the waste of materials, time and costs during IPh. Table 1 illustrates proportionality between study objectives and questionnaire content.

**Table 1.** Questionnaire content.

| Subsection | Variables | Objective |
|---|---|---|
| **Section I: Profile of Respondent** | | |
| Respondent Organization | Owner-donor-consultant-contractor | Study the relations based on characteristic of organization between commitment to SS and minimizing CW. |
| | Company classification. | |
| | Numbers and value of CPs | |
| Respondent personality | Respondent position: chairman-general manager-projects managers-project manager-site engineer-office engineer- other | Study the relations based on specification of the participant between commitment to SS and minimizing CW. |
| | Respondent qualification, classification and experience | |
| **Section II: Safety management practice in construction projects** | | |
| Safety management practices in the institution | Data record, Safety plan, Safety producers, Safety training, Law and regulation of safety | To highlight the safety management practices in construction institutions |
| Safety factors have positive impacts on minimizing the CW | Commitment degree to SS | - To determine the commitment degree of safety factors on CPs then ranking it according its RII. |
| | Safety factors have positive effect on waste in CPs during IPh | - To determine the effect of each safety factors on minimizing waste in CPs, then ranking it according its RII. |
| **Section III:** Respondent recommendations to minimizing CW by using SS in CPs | | |

### 3.2. Structured Interviews and Pilot Study

To revise the draft questionnaire, the research team performed structured interviews with engineers and managers working for consulting offices, construction contracting companies, donors and public owners. The questions were presented in the same wording and order to all interviewees. In addition, observations on workplace conditions, notes of inspectors and project documents, such as daily reports, supervisors' instructions, working original and shop drawings and progress reports, were considered. To test the normality, validity and reliability of the scales used for some of the questions, a pilot study was conducted following two distinct procedures. In the first procedure, 15 experts, project managers and engineers from different contracting companies were interviewed face-to-face. In the second procedure, 15 experts with more than 10 years of experience in CPs were invited to review the draft questionnaire. Some of the invited experts were academicians, whereas others were professionals. The interviews and pilot study helped identify the potential problems and errors in the draft questionnaire. In addition, the wording of numerous questions was improved to enhance understanding and avoid misinterpretation and/or possible different readings of the same question. The interviews and pilot study were also helpful in filtering safety factors with a positive effect on minimising waste in CPs during IPh. The professionals were asked to provide their opinions regarding the factors found in the literature and were welcome to add other possible factors on the basis of their experience. All collected information was synthesised into a final version of the questionnaire, which was then distributed to the target group for this research, as presented in the next section.

### 3.3. Research Population and Sample Size

The target group for this research included consulting offices, contracting companies, owner agencies and donor agencies. Only contracting companies registered in the country's union of contactors and classified by the 'National Classification Committee' as 'first class' with valid registration were approached in this research [28]. Other companies were excluded due to their low CS and waste management practices and limited administration experience. A total of 66 active companies in the country met the research's target criteria. For the consulting offices, 68 firms registered in the country's engineer syndicate were targeted. Meanwhile, 15 owner agencies, which consisted of ministries, municipalities, international agencies, nongovernmental organisations and public project owners, were included. Ten active donor agencies were also contacted. Equation (1) was used to estimate the total sample size required for this research, and Equation (2) was applied to correct the outcome of Equation 1 for the finite population [62]:

$$S = \frac{Z^2 \times P \times (1 - P)}{C^2}, \tag{1}$$

$$S_{new} = \frac{SS}{1 + \frac{SS-1}{pop}}, \tag{2}$$

where $S$ is the sample size; $Z$ denotes the Z-value from the normal distribution table, which is set as 1.96 and corresponds to a 95% confidence interval; $P$ represents the percentage probability of making a decision, which is expressed as a decimal (assumed to be 0.50 in this study); and $C$ refers to the maximum error of estimation (assumed to be 0.08 in this study). A population size of 150 was obtained using Equation (1) with the assumed values. This number was reduced using Equation (2), and Figure 4 shows the results for different types of contacted companies and agencies. This figure also presents the number of returned questionnaires with the total percentage of responses. A total of 111 filled questionnaires were returned to the research team for response analysis.

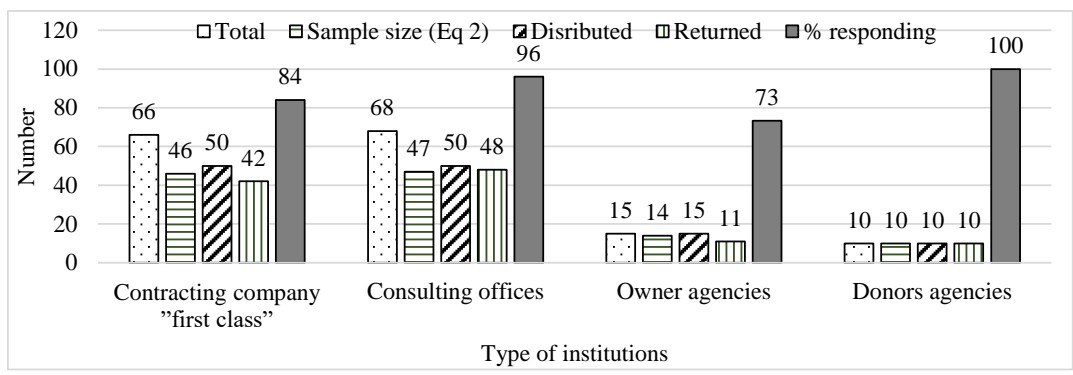

**Figure 4.** Statistics regarding the research's target groups.

### 3.4. S Factors during IPh and Their Effect on Minimising CW in CPs

From the literature review, structured interviews and pilot study, 24 SS factors during IPh were identified as exerting a potential effect on minimising CW. Table 2 lists these factors and indicates the number of paragraphs used under each evaluated factor. For example, Table 3 shows the paragraphs considered for F1 (i.e., appropriate scaffolding work for SS).

**Table 2.** Safety factors during IPh identified as having potential positive effects on minimising CW.

| # | Factor | Number of Paragraphs |
|---|--------|---------------------|
| F1 | Appropriate scaffolding work for SS | 7 |
| F2 | Appropriate mobile scaffolds for SS | 9 |
| F3 | Appropriate ladders to reach high areas for SS | 4 |
| F4 | Appropriate roof work for SS | 6 |
| F5 | Appropriate access workplace for SS | 5 |
| F6 | Housekeeping | 3 |
| F7 | Personal protective equipment | 8 |
| F8 | Site safety information documents | 9 |
| F9 | Safety action plan | 9 |
| F10 | Competency of workers and ongoing training | 5 |
| F11 | Monitoring system | 10 |
| F12 | Risk reports | 2 |
| F13 | Accident reports | 3 |
| F14 | Discipline | 4 |
| F15 | Inspections | 2 |
| F16 | Communication in the workplace | 4 |
| F17 | Responsibility for safety in the workplace | 4 |
| F18 | Cooperation | 2 |
| F19 | Handling | 6 |
| F20 | Workers | 15 |
| F21 | Management | 10 |
| F22 | Site condition | 7 |
| F23 | Procurement | 7 |
| F24 | External factors | 6 |

**Table 3.** Paragraphs used for F1 (appropriate scaffolding work for SS).

| # | Paragraph |
|---|---|
| 1. | Adoption of executive plan of scaffolding works in accordance with the safety standards before starting scaffolding work |
| 2. | Proper installation of scaffolding (scaffolding is placed on sound footing, braced and tied properly, with toe boards in place) |
| 3. | Using metal sheet from full panels (non-fragmented) to install the base of the scaffolding and supporting these plates in a strong and safe way |
| 4. | Providing the scaffolding with an access ladder |
| 5. | Installing handrails and mid-rails (side protections) in the needed places for scaffolding |
| 6. | Using scaffolding trestles properly and safely |
| 7. | Selecting platelet (ground) scaffolding to bear potential weights loaded on them |

*3.5. Data Measurement and Analysis*

The questionnaire begins with a covering letter. Respondents were requested to answer questions honestly and confidentially. Many calls and visits were conducted to encourage them and to facilitate and overcome any problems.

In this research, ordinal scales (1,2,3,4,5,6,7,8,9,10) were used. Ordinal scale is a ranking or a rating data that normally uses integers in ascending or descending order.

The collected data were first sorted, edited, coded and then analysed using descriptive and inferential statistical tools. All questionnaire results were inputted into IBM SPSS Statistics (version 22). Nine types of data analysis techniques were used in this study, as follows:

1. Frequency and descriptive analyses,
2. Cronbach's alpha and split half (Spearman–Brown) for reliability statistics,
3. Pearson's correlation coefficients for validity,
4. Kolmogorov–Smirnov (KS) test for normality distribution,
5. One-sample *t*-test to determine if the null hypothesis that the mean of a distribution is equal to a certain value is supported,
6. Independent-sample *t*-test to examine if a statistically significant difference exists in rank mean between two groups,
7. ANOVA to check for any significant difference between more than two groups,
8. Linear regression model to relate safety factors to CW,
9. Effect size to measure the strength of the relationship between two variables on a numeric scale.

All of the aforementioned tools are typical statistical devices, and readers can use any available statistics books, such as Berger's (2002) and the Probability and Statistics Cookbook [49,50], for additional information. Moreover, the relative importance index (RII) was used to rank the questionnaire factors [63–69]. RII was computed using Equation (3) [70,71]:

$$\mathrm{RII} = \frac{\sum W}{A \times N} \times 100\%,\tag{3}$$

where W is the weight given to each factor by the respondents, A indicates the highest weight (10 in this study) and N represents the total number of respondents. The RII value ranges from 0% (not inclusive) to 100%; the higher the RII value, the greater the attribute effect. However, RII does not reflect the relationship amongst various attributes.

## 4. Findings and Discussion

### 4.1. General Information about the Institutions and Participants

Figure 5a shows the distribution of the institutions that participated in this study. Contracting companies and consulting offices represent most of the contacted establishments with a total percentage of 81.1%. The 42 selected contracting companies are classified as 'first class' in construction building projects and are involved in other types of construction, such as roads and sewage systems. Figure 5b shows the types of projects in which these contracting companies are involved. The sample includes all parties that are directly related to the design process: the contractor as the executor of the design, the consultant as the designer and supervisor and the owner as the beneficiary and financier. Therefore, the opinions of all parties involved in a CP were collected in this study.

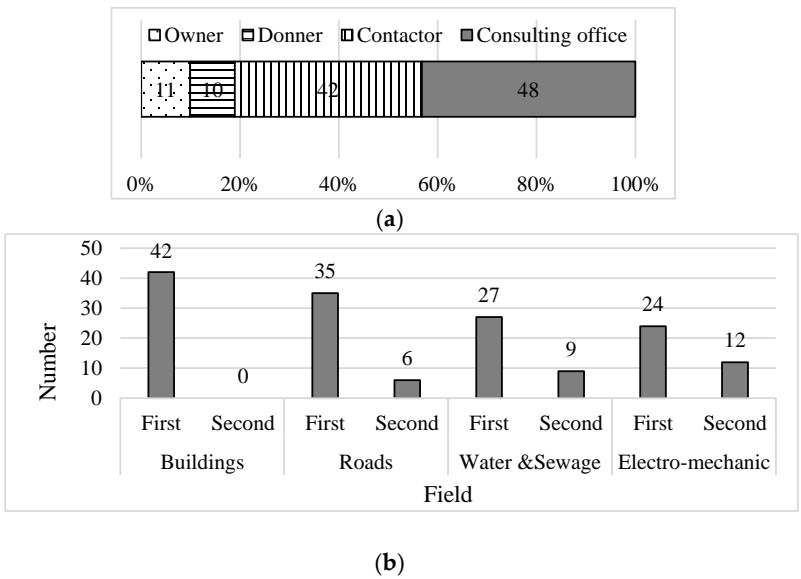

(a)

(b)

**Figure 5.** Information about the responding institutions: (**a**) type of establishments and (**b**) fields of contracting companies.

Civil engineers constitute 90 of the 111 questionnaire respondents. Most of the respondents (57%) hold a master's degree. The majority of the participants (94%) have more than five years of experience in CPs and 34 have more than 15 years of experience. In addition, 67% and 61% of the participants have more than five years of experience in CS and CWM, respectively. Moreover, 83% and 68% of the respondents have at least one course in CS and CWM, respectively. Five (4.5%), 17 (15.3%), 29 (26.1%), 22 (19.8%), 30 (27.1%), five (4.5%) and three (2.7%) are chairmen, general managers, projects managers, project managers, site engineers, office engineers and others, respectively. Furthermore, the experts who participated in this research exhibit academic, practical, cultural and scientific diversities. The respondents have studied and worked in engineering in several (developing and developed) countries; the responding institutions (national and international) have finished several CPs in numerous countries, thereby providing a universal aspect to the results of this study.

### 4.2. General Findings about the Questionnaire Questions

The KS test of normality resulted in *p*-values greater than the 0.05 significance level, which indicates that each field of IPh in a CP is normally distributed. Table 4 presents the results of this test for some fields used in the questionnaire.

**Table 4.** Results of the KS test of normality.

| Field | | Statistic | df * | *p*-Value |
|---|---|---|---|---|
| **Safety Management Practice in CPs** | | 0.054 | 111 | 0.200 |
| Safety factors related to positive impacts on minimizing the CW during IPh | Commitment degree | 0.081 | 111 | 0.073 |
| | Waste in Materials | 0.078 | 111 | 0.094 |
| | Time overrun | 0.060 | 111 | 0.200 |
| | Cost overrun | 0.063 | 111 | 0.200 |
| CWM in CPs | | 0.084 | 111 | 0.051 |
| Degree of commitment to minimize waste of time, material, and cost during IPh | | 0.070 | 111 | 0.171 |

* degree of freedom.

The internal and structure validities of the questionnaire were tested using Pearson's correlation analysis, which measures the correlation coefficient (R) between each paragraph in one field and the entire field and between each field and the validity of the entire questionnaire. This test measures the R between one field and all fields of the questionnaire with the same level of scales. The test indicated that the R of each paragraph of safety factors during IPh is significant at $\alpha = 0.05$; thus, the paragraphs of this field are consistent and valid for measuring the value for which it is set. Table 5 presents the test results of the first two considered factors and their respective paragraphs

**Table 5.** R of each paragraph of IPh and the total of this field.

| No. | Commitment Degree | | Waste in Material | | Time Overrun | | Cost Overrun | |
|---|---|---|---|---|---|---|---|---|
| | R | *p*-Value (Sig.) | R | *p*-Value (Sig.) | R | *p*-Value (Sig.) | R | *p*-Value (Sig.) |
| **F1. Appropriate Scaffolding work for the SS** | | | | | | | | |
| 1. | 0.756 | 0.000 ** | 0.526 | 0.003 ** | 0.595 | 0.001 ** | 0.614 | 0.000 ** |
| 2. | 0.854 | 0.000 ** | 0.830 | 0.000 ** | 0.519 | 0.003 ** | 0.797 | 0.000 ** |
| 3. | 0.824 | 0.000 ** | 0.475 | 0.008 ** | 0.735 | 0.000 ** | 0.610 | 0.000 ** |
| 4. | 0.853 | 0.000 ** | 0.710 | 0.000 ** | 0.791 | 0.000 ** | 0.602 | 0.000 ** |
| 5. | 0.791 | 0.000 ** | 0.744 | 0.000 ** | 0.772 | 0.000 ** | 0.561 | 0.001 ** |
| 6. | 0.601 | 0.000 ** | 0.737 | 0.000 ** | 0.777 | 0.000 ** | 0.737 | 0.000 ** |
| 7. | 0.623 | 0.000 ** | 0.691 | 0.000 ** | 0.675 | 0.000 ** | 0.721 | 0.000 ** |
| **F2. Appropriate Mobile Scaffolds for the SS** | | | | | | | | |
| 1. | 0.640 | 0.000 ** | 0.698 | 0.000 ** | 0.678 | 0.000 ** | 0.693 | 0.000 ** |
| 2. | 0.831 | 0.000 ** | 0.833 | 0.000 ** | 0.798 | 0.000 ** | 0.639 | 0.000 ** |
| 3. | 0.830 | 0.000 ** | 0.894 | 0.000 ** | 0.892 | 0.000 ** | 0.814 | 0.000 ** |
| 4. | 0.837 | 0.000 ** | 0.955 | 0.000 ** | 0.943 | 0.000 ** | 0.809 | 0.000 ** |
| 5. | 0.872 | 0.000 ** | 0.863 | 0.000 ** | 0.963 | 0.000 ** | 0.896 | 0.000 ** |
| 6. | 0.384 | 0.036 * | 0.593 | 0.001 ** | 0.493 | 0.006 ** | 0.592 | 0.001 ** |
| 7. | 0.440 | 0.016 * | 0.590 | 0.001 ** | 0.437 | 0.016 * | 0.462 | 0.010 * |
| 8. | 0.670 | 0.000 ** | 0.720 | 0.000 ** | 0.773 | 0.000 ** | 0.504 | 0.004 ** |
| 9. | 0.944 | 0.000 ** | 0.832 | 0.000 ** | 0.907 | 0.000 ** | 0.730 | 0.000 ** |

* Correlation is significant at the 0.05 level, ** Correlation is significant at the 0.01 level.

*4.3. Testing of Hypotheses*

With reference to previous studies as in the literature review and Figure 2, safety elements in CPs around the world have been assembled. These factors were studied from the point of view of its impact on CW. The collected factors of SS were presented and discussed with experienced construction managers. The researcher added several other factors of SS, which weren't on lists in previous studies derived from the experiences of the researcher and the experts who were interviewed during this study, or even at the stage of pilot study. To find the relationship between degree of commitment to SS and CW through project cycle, three hypotheses were tested in this study, as follows:

1.　'An inverse relationship, which is statistically significant at $\alpha = 0.05$, exists between commitment to SS and non-physical waste (time overrun) in CP'.
2.　'An inverse relationship, which is statistically significant at $\alpha = 0.05$, exists between commitment to SS and non-physical waste (cost overrun) in CP'.
3.　'An inverse relationship, which is statistically significant at $\alpha = 0.05$, exists between commitment to SS and material overrun (physical waste) in CP'.

Parametric tests were performed to determine if the hypotheses were supported. For example, the *t*-test and ANOVA were used to conduct the analysis. One sample test was used to verify whether the population mean is equal to the midpoint (6) in the Likert scale. These tests are appropriate for ordinal and numerical data. For the alternative hypothesis (H1), the average degree is not equal to 6.

If the *p*-value is greater than the significance level $\alpha = 0.05$, then the null hypothesis is not rejected (the average response to the phenomenon under study does not differ significantly from the degree of neutrality, i.e., 6). If the calculated *p*-value is smaller than the significance level $\alpha = 0.05$, then the null hypothesis is rejected; that is, the average differs from 6. In this case, the sign of the statistics test indicates how different the mean respondents are from 6. A positive sign indicates that the average is greater than 6, whereas a negative sign shows that the average is smaller than 6.

The output of these tests supports all hypotheses; hence, an inverse relationship that is statistically significant at $\alpha \leq 0.05$ exists between commitment to the design for SS during IPh and waste (materials, time and cost) in CP.

*4.4. Main Factors of SS with Positive Effects on Minimising CW during IPh*

Table 6 summarises the main safety factors with positive effects on minimising the waste of materials, time and cost during IPh. The highest ranked factor for minimising waste in materials and time is 'appropriate handling for SS', whereas that for minimising waste in cost is 'appropriate management for SS'. The lowest ranked factor for minimising waste in materials is 'monitoring system for SS', whereas that for minimising waste in time is 'accident reports'. Finally, the lowest ranked factor for minimising waste in cost is 'competency of workers and ongoing training'. The respondents agreed to these factors, and the sign of the test is positive (RII > 60%). Table 6 illustrates the RII and rank of the safety factors during IPh.

**Table 6.** RII and rank of the safety factors during IPh.

| Factor | Commitment to | | | Material | | | Time | | | Cost | | |
|---|---|---|---|---|---|---|---|---|---|---|---|---|
| | RII | *p*-Value (Sig.) | ES | RII | *p*-Value (Sig.) | ES | RII | *p*-Value (Sig.) | ES | RII | *p*-Value (Sig.) | ES |
| F1 | 78.79 | <0.001 | 1.29 | 78.09 | <0.001 | 1.48 | 79.64 | <0.001 | 1.7 | 79.26 | <0.001 | 1.6 |
| F2 | 81.23 | <0.001 | 1.72 | 76.51 | <0.001 | 1.14 | 77.53 | <0.001 | 1.31 | 77.14 | <0.001 | 1.31 |
| F3 | 79.45 | <0.001 | 1.36 | 76.39 | <0.001 | 1.03 | 78.55 | <0.001 | 1.36 | 76.39 | <0.001 | 1.13 |
| F4 | 75.72 | <0.001 | 0.96 | 75.42 | <0.001 | 1.18 | 76.21 | <0.001 | 1.44 | 76.59 | <0.001 | 1.33 |
| F5 | 80.63 | <0.001 | 1.66 | 73.58 | <0.001 | 0.76 | 76.68 | <0.001 | 1.36 | 75.47 | <0.001 | 0.96 |
| F6 | 82.34 | <0.001 | 2.17 | 74.73 | <0.001 | 0.88 | 77.95 | <0.001 | 1.37 | 78.28 | <0.001 | 1.43 |
| F7 | 76.66 | <0.001 | 1.21 | 72.45 | <0.001 | 0.82 | 73.33 | <0.001 | 1.03 | 72.78 | <0.001 | 0.83 |
| F8 | 73.03 | <0.001 | 0.8 | 75.29 | <0.001 | 1.16 | 76.3 | <0.001 | 1.46 | 76.96 | <0.001 | 1.76 |
| F9 | 72.78 | <0.001 | 0.75 | 73.04 | <0.001 | 0.88 | 73.02 | <0.001 | 0.99 | 73.12 | <0.001 | 0.98 |
| F10 | 67.87 | <0.001 | 0.4 | 75.87 | <0.001 | 0.49 | 73.13 | <0.001 | 0.89 | 72.07 | <0.001 | 0.83 |
| F11 | 72.39 | <0.001 | 0.720. | 71.95 | <0.001 | 0.83 | 72.67 | <0.001 | 1.02 | 73.04 | <0.001 | 0.99 |
| F12 | 75.54 | <0.001 | 0.88 | 72.16 | <0.001 | 0.7 | 74.09 | <0.001 | 0.88 | 75.81 | <0.001 | 1.07 |
| F13 | 73.57 | <0.001 | 0.75 | 72.37 | <0.001 | 0.73 | 72.64 | <0.001 | 0.81 | 74.02 | <0.001 | 0.97 |
| F14 | 77.25 | <0.001 | 1.29 | 74.45 | <0.001 | 0.96 | 74.82 | <0.001 | 1.12 | 76.66 | <0.001 | 1.39 |
| F15 | 74.05 | <0.001 | 0.81 | 75.49 | <0.001 | 1 | 75.99 | <0.001 | 1.13 | 77.16 | <0.001 | 1.09 |
| F16 | 75.54 | <0.001 | 0.57 | 73.37 | <0.001 | 0.86 | 75.11 | <0.001 | 1.13 | 75.49 | <0.001 | 1.12 |
| F17 | 73.51 | <0.001 | 0.73 | 77.23 | <0.001 | 1.28 | 78.38 | <0.001 | 1.53 | 79.05 | <0.001 | 1.58 |
| F18 | 72.11 | <0.001 | 0.67 | 75.94 | <0.001 | 0.93 | 78.19 | <0.001 | 1.3 | 76.66 | <0.001 | 1.07 |
| F19 | 77.35 | <0.001 | 1.17 | 81.54 | <0.001 | 1.75 | 81.44 | <0.001 | 1.9 | 81.6 | <0.001 | 1.83 |
| F20 | 79.63 | <0.001 | 1.42 | 79.44 | <0.001 | 1.6 | 79.15 | <0.001 | 1.65 | 81.39 | <0.001 | 1.82 |
| F21 | 81.63 | <0.001 | 1.56 | 80.03 | <0.001 | 1.63 | 81.04 | <0.001 | 1.97 | 82.55 | <0.001 | 2.01 |
| F22 | 79.56 | <0.001 | 1.39 | 78.35 | <0.001 | 1.49 | 78.71 | <0.001 | 1.55 | 79.08 | <0.001 | 1.48 |
| F23 | 79.35 | <0.001 | 1.25 | 78.97 | <0.001 | 1.57 | 79.85 | <0.001 | 1.71 | 81.14 | <0.001 | 1.75 |
| F24 | 80.88 | <0.001 | 1.46 | 79.71 | <0.001 | 1.52 | 80.61 | <0.001 | 1.81 | 81.2 | <0.001 | 1.83 |

Factor no. 19, i.e. 'handling', ranked highest for minimising waste in materials, with RII = 81.54% and *p*-value < 0.001. This factor ranked highest for minimising waste in time and second for minimising cost overrun. This result agrees with the findings of previous studies [10,36,61], which confirm that the cost of construction materials may be up to 65% of the total cost incurred in the construction of a civil engineering structure. However, such cost is dependent upon the type of project and the construction technique and plant used [64]. Therefore, the main objective of material management and planning is to supply the right construction materials in the right place and the right quantities when needed.

Factor no. 21, namely, 'management', ranked second for minimising waste (materials and time), with RII = 80.0% and 81.0%, respectively, and *p*-value < 0.001. This factor ranked highest in minimising cost. This result agrees with the findings of previous studies [18,36]. The importance of this factor in reducing waste is highlighted through its association with several aspects, such as good project organising and monitoring, powerful site management, selecting supervisors with good and strong experience and avoiding inappropriate construction methods. Appropriate planning and construction management substantially reduce the wastage of construction materials. This case in turn improves or increases the performance and economy of the organisation. Poor construction progress may be generally due to poor planning and management of construction material. The management should

be focussed on organising, procuring, sorting and distributing construction materials at appropriate times and places.

Factor no. 24, namely, 'external factors', ranked third for minimising waste (materials and time), with RII = 79.71% and 80.61%, respectively, and *p*-value < 0.001. This factor ranked fourth for minimising cost overrun. This result agrees with the findings of previous studies [13]. The importance of this factor lies in its containment: (1) avoiding negative weather effect, (2) accidents, (3) vandalism and (4) damages caused by third parties, (5) compliance with laws and regulations and (6) capability to predict local conditions.

Factor no. 20, namely, 'workers', ranked fourth for minimising waste in materials, with RII = 79.44% and *p*-value < 0.001. This factor ranked sixth for minimising waste in time and fourth in minimising cost overrun. This result agrees with the findings of previous studies [60]. The importance of this factor lies in its containment: (1) preventing working errors during construction, (2) selecting and providing skilled and experienced workers, (3) avoiding the bad behaviour of workers, (4) reducing the damage caused by workers, (5) adequate and well-trained workers, (6) quality assurance, (7) increasing the enthusiasm of workers, (8) avoiding inappropriate use of materials by workers, (9) good documentation of stored materials, (10) requiring workers to wear protective clothing, (11) increasing awareness of the workers, (12) avoiding overtime for workers, (13) providing breaks for workers, (14) providing insurance policy for workers throughout the project duration and (15) appropriate salary based on the nature and number of working hours.

Companies with a waste management culture within the organisation invest in CWM by employing waste management workers, purchasing equipment and/or machines for waste minimisation and improving workers' skills.

Factor no. 23, namely, 'procurement', ranked fifth for minimising waste in materials, with RII = 78.97% and 78.71% and *p*-value < 0.001. This factor ranked sixth for minimising waste in time and fourth for minimising cost overrun. This result agrees with the findings of previous studies [12,72,73]. The importance of this factor lies in its containment: (1) preventing mistakes in supplies, (2) avoiding transport error and reducing supplier errors, (3) preventing mistakes in quantity surveys, (4) avoiding incorrect procedures of material delivery, (5) avoiding increase over the allocated quantities to purchase, (6) reducing the repetition of change orders and (7) reducing the waiting time for equipment replacement. Material procurement and storage on construction sites must be properly managed, planned and executed to avoid the negative effects of material on environments and shortage or excessive material inventory on construction site and the deficiencies in the supply and flow of construction materials.

Factor no. 22, namely, 'site condition', ranked sixth for minimising waste in materials, with RII = 79.35% and *p*-value < 0.001. This factor ranked seventh for minimising waste in time and for minimising cost overrun. This result agrees with the findings of previous studies (10, 11). The importance of this factor lies in its containment: (1) reducing the remaining materials at the site, (2) reducing waste in the site, (3) avoiding congestion and overcrowding, (4) avoiding lighting problem, (5) facilitating the access to construction sites, (6) considering non-visual ground conditions and (7) avoiding interference of any other crews in the site.

Factor no. 1, namely, 'appropriate scaffolding work for SS', ranked seventh for minimising waste in materials, with RII = 78.09% and *p*-value < 0.001. This factor ranked fifth for minimising waste in time and sixth for minimising cost overrun. This result agrees with [35]. The importance of this factor lies in its containment: (1) adopting an executive plan of scaffolding works in accordance with the safety standards before starting scaffolding work, (2) properly installing scaffolding (scaffolding is placed on sound footing, braced and tied properly, with toe boards in place), (3) using a metal sheet from full panels (non-fragmented) to install the scaffolding base and supporting these plates in a strong and safe way, (4) providing scaffolding with an access ladder, (5) installing handrails and mid-rails (side protections) in the needed places for scaffolding, (6) using scaffolding trestles properly and safely and (7) selecting platelet (ground) scaffolding to bear potential weights loaded on them.

Factor no. 11, namely, 'monitoring system', ranked lowest for minimising waste in materials, with RII = 71.95% and *p*-value < 0.001. Factor no. 13, namely, 'accident reports', ranked lowest for minimising waste in time, with RII = 72. 64% and *p*-value < 0.001. Factor no. 10, namely, 'competency of workers and ongoing training', ranked lowest for minimising cost overrun, with RII = 72.07% and *p*-value < 0.001. The explanation for this result of these factors is related to post-events and not pre-events.

*4.5. Prediction Equations*

The best linear models related to the variables (commitment to SS during IPh and minimising waste, namely, materials, time and cost) in CPs were developed on the basis of the questionnaire results. Figure 6 shows the results obtained for 'minimising waste of materials, time and cost' as a function of 'commitment to SS'.

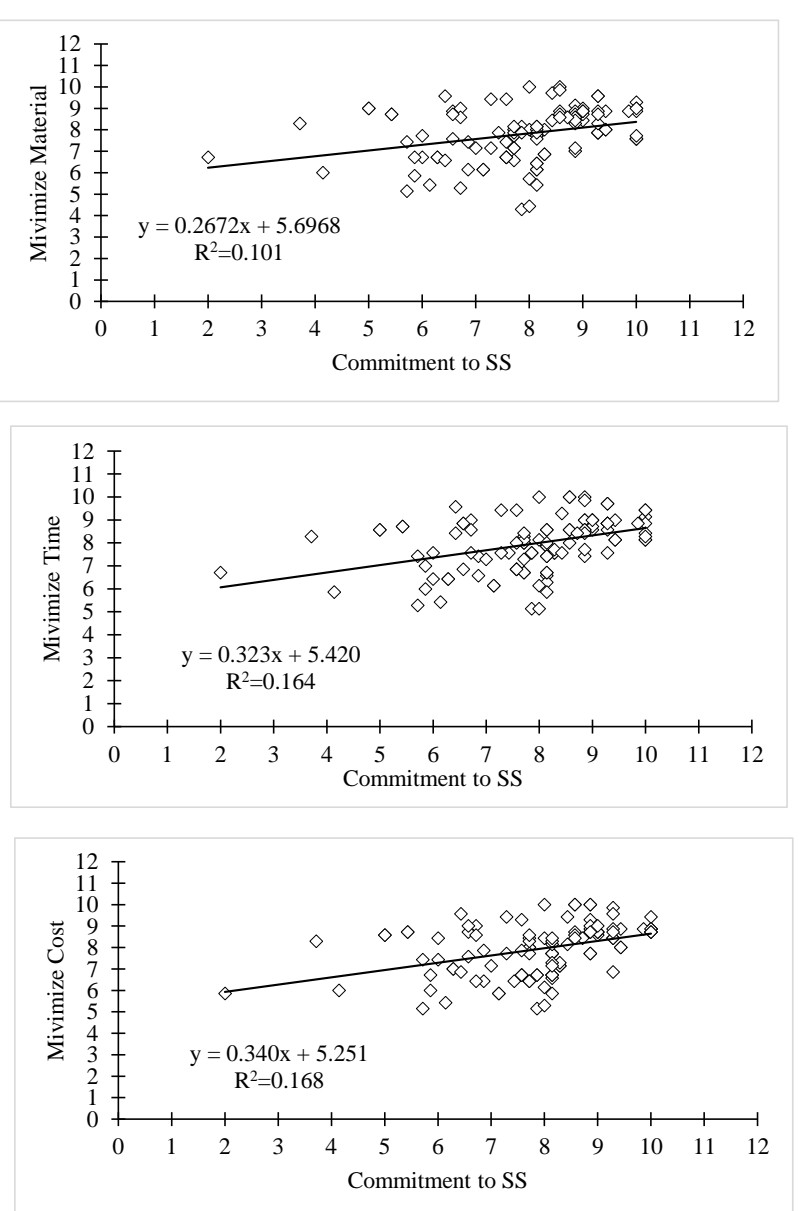

**Figure 6.** Appropriate scaffolding work for SS.

The obtained equations, as shown in Table 7, are used as predictive equations to minimise waste (materials, time and cost) on the basis of the commitment degree to each factor of SS during

IPh. A statistically significant relationship ($\alpha < 0.05$) is observed between commitment to SS and minimising CW (in materials, time and cost) during IPh. For example, the relationships between the appropriate scaffolding work for SS and minimising CW are 0.318, 0.406 and 0.410 for materials, time and cost, respectively. Thus, a positive relationship exists between commitment to SS and minimising CW. The determination coefficients, $R^2$, are equal to 0.10, 0.16 and 0.17 for materials, time and cost, respectively. These values indicate that 10.0%, 16.0% and 17.0% of the variabilities of commitment to the appropriate scaffolding work for SS are meant to minimise CW in materials, time and cost, respectively. The *p*-values are less than 5%; thus, reducing CW has a significant positive effect on the degree of commitment to SS. Table 6 presents the prediction equations related to commitment for all factors studied in this research with the three types of CW (materials, time and cost).

**Table 7.** Predictive equations to minimize waste (materials, time and cost) according to the degree of commitment to each factor of SS during IPH.

| Commitment | Material | | | | Time | | | | Cost | | | |
|---|---|---|---|---|---|---|---|---|---|---|---|---|
| Appropriate scaffolding work for SS | $r$ | 0.318 | $r^2$ | 0.101 | $r$ | 0.406 | $r^2$ | 0.164 | $r$ | 0.410 | $r^2$ | 0.168 |
| | $y = 5.697 + 0.267x$ std 0.610  0.076 | | | | $y = 5.420 + 0.323x$ std 0.558  0.070 | | | | $y = 5.251 + 0.340x$ std 0.579  0.072 | | | |
| Appropriate mobile scaffolds for SS | $r$ | 0.458 | $r^2$ | 0.210 | $r$ | 0.552 | $r^2$ | 0.305 | $r$ | 0.568 | $r^2$ | 0.322 |
| | $y = 3.273 + 0.539x$ std 0.823  0.100 | | | | $y = 2.870 + 0.601x$ std 0.714  0.087 | | | | $y = 2.829 + 0.601x$ std 0.687  0.084 | | | |
| Appropriate ladders to reach high areas for SS | $r$ | 0.576 | $r^2$ | 0.332 | $r$ | 0.520 | $r^2$ | 0.270 | $r$ | 0.534 | $r^2$ | 0.285 |
| | $y = 2.533 + 0.643x$ std 0.705  0.087 | | | | $y = 3.899 + 0.498x$ std 0.633  0.078 | | | | $y = 3.314 + 0.544x$ std 0.667  0.083 | | | |
| Appropriate roof work for SS | $r$ | 0.312 | $r^2$ | 0.097 | $r$ | 0.352 | $r^2$ | 0.124 | $r$ | 0.335 | $r^2$ | 0.13 |
| | $y = 5.654 + 0.249x$ std 0.564  0.073 | | | | $y = 5.771 + 0.244x$ std 0.481  0.062 | | | | $y = 5.718 + 0.256x$ std 0.534  0.069 | | | |
| Appropriate access work place for SS | $r$ | 0.365 | $r^2$ | 0.133 | $r$ | 0.499 | $r^2$ | 0.249 | $r$ | 0.375 | $r^2$ | 0.141 |
| | $y = 3.138 + 0.523x$ std 1.043  0.128 | | | | $y = 3.679 + 0.495x$ std 0.672  0.082 | | | | $y = 3.608 + 0.489x$ std 0.943  0.116 | | | |
| House-keeping | $r$ | 0.264 | $r^2$ | 0.070 | $r$ | 0.406 | $r^2$ | 0.164 | $r$ | 0.363 | $r^2$ | 0.132 |
| | $y = 3.949 + 0.428x$ std 1.242  0.150 | | | | $y = 3.679 + 0.495x$ std 0.672  0.082 | | | | $y = 3.608 + 0.489x$ std 0.943  0.116 | | | |
| Personal protective equipment | $r$ | 0.233 | $r^2$ | 0.054 | $r$ | 0.234 | $r^2$ | 0.055 | $r$ | 0.261 | $r^2$ | 0.068 |
| | $y = 5.274 + 0.257x$ std 0.801  0.103 | | | | $y = 5.652 + 0.219x$ std 0.680  0.087 | | | | $y = 5.049 + 0.291x$ std 0.801  0.103 | | | |
| Site safety information documents | $r$ | 0.204 | $r^2$ | 0.042 | $r$ | 0.089 | $r^2$ | 0.008 | $r$ | 0.180 | $r^2$ | 0.035 |
| | $y = 6.319 + 0.166x$ std 0.570  0.076 | | | | $y = 7.184 + 0.061x$ std 0.493  0.066 | | | | $y = 6.876 + 0.112x$ std 0.420  0.056 | | | |
| Safety action plan | $r$ | 0.291 | $r^2$ | 0.085 | $r$ | 0.263 | $r^2$ | 0.069 | $r$ | 0.241 | $r^2$ | 0.058 |
| | $y = 5.454 + 0.254x$ std 0.598  0.080 | | | | $y = 5.819 + 0.204x$ std 0.536  0.072 | | | | $y = 5.925 + 0.191x$ std 0.549  0.073 | | | |
| Competency of workers and ongoing training | $r$ | 0.234 | $r^2$ | 0.055 | $r$ | 0.405 | $r^2$ | 0.164 | $r$ | 0.362 | $r^2$ | 0.131 |
| | $y = 4.966 + 0.386x$ std 1.086  0.154 | | | | $y = 5.241 + 0.305x$ std 0.466  0.066 | | | | $y = 5.392 + 0.267x$ std 0.465  0.066 | | | |
| Monitoring system | $r$ | 0.335 | $r^2$ | 0.112 | $r$ | 0.266 | $r^2$ | 0.071 | $r$ | 0.345 | $r^2$ | 0.119 |
| | $y = 5.157 + 0.282x$ std 0.564  0.076 | | | | $y = 5.874 + 0.192x$ std 0.496  0.067 | | | | $y = 5.388 + 0.265x$ std 0.513  0.069 | | | |
| Risk reports | $r$ | 0.335 | $r^2$ | 0.112 | $r$ | 0.315 | $r^2$ | 0.099 | $r$ | 0.263 | $r^2$ | 0.069 |
| | $y = 4.728 + 0.329x$ std 0.687  0.089 | | | | $y = 5.264 + 0.284x$ std 0.636  0.082 | | | | $y = 5.923 + 0.219x$ std 0.597  0.077 | | | |
| Accident reports | $r$ | 0.451 | $r^2$ | 0.203 | $r$ | 0.386 | $r^2$ | 0.149 | $r$ | 0.372 | $r^2$ | 0.138 |
| | $y = 4.138 + 0.421x$ std 0.605  0.080 | | | | $y = 4.793 + 0.336x$ std 0.538  0.077 | | | | $y = 4.936 + 0.300x$ std 0.543  0.072 | | | |
| Discipline | $r$ | 0.451 | $r^2$ | 0.203 | $r$ | 0.386 | $r^2$ | 0.149 | $r$ | 0.372 | $r^2$ | 0.138 |
| | $y = 4.724 + 0.350x$ std 0.605  0.080 | | | | $y = 4.689 + 0.361x$ std 0.692  0.088 | | | | $y = 5.194 + 0.353x$ std 0.620  0.079 | | | |

**Table 7.** *Cont.*

| Commitment | Material | | | | Time | | | | Cost | | | |
|---|---|---|---|---|---|---|---|---|---|---|---|---|
| Inspections | $r$ | 0.310 | $r^2$ | 0.096 | $r$ | 0.365 | $r^2$ | 0.133 | $r$ | 0.394 | $r^2$ | 0.155 |
| | $y = 4.976 + 0.348x$ std 0.600　0.079 | | | | $y = 5.526 + 0.280x$ std 0.558　0.073 | | | | $y = 5.289 + 0.328x$ std 0.616　0.081 | | | |
| Communication in the workplace | $r$ | 0.358 | $r^2$ | 0.128 | $r$ | 0.374 | $r^2$ | 0.140 | $r$ | 0.197 | $r^2$ | 0.039 |
| | $y = 5.790 + 0.205x$ std 0.411　0.051 | | | | $y = 6.124 + 0.184x$ std 0.350　0.044 | | | | $y = 6.797 + 0.100x$ std 0.382　0.048 | | | |
| Responsibility for safety in the workplace | $r$ | 0.288 | $r^2$ | 0.083 | $r$ | 0.163 | $r^2$ | 0.027 | $r$ | 0.210 | $r^2$ | 0.044 |
| | $y = 6.190 + 0.209x$ std 0.504　0.067 | | | | $y = 7.061 + 0.106x$ std 0.465　0.061 | | | | $y = 6.903 + 0.136x$ std 0.460　0.061 | | | |
| Cooperation | $r$ | 0.303 | $r^2$ | 0.092 | $r$ | 0.395 | $r^2$ | 0.156 | $r$ | 0.453 | $r^2$ | 0.205 |
| | $y = 5.529 + 0.286x$ std 0.641　0.086 | | | | $y = 5.614 + 0.306x$ std 0.506　0.068 | | | | $y = 4.861 + 0.389x$ std 0.545　0.073 | | | |
| Handling | $r$ | 0.598 | $r^2$ | 0.358 | $r$ | 0.453 | $r^2$ | 0.205 | $r$ | 0.330 | $r^2$ | 0.109 |
| | $y = 4.301 + 0.498x$ std 0.503　0.064 | | | | $y = 5.473 + 0.345x$ std 0.513　0.065 | | | | $y = 6.125 + 0.263x$ std 0.569　0.072 | | | |
| Workers | $r$ | 0.597 | $r^2$ | 0.357 | $r$ | 0.572 | $r^2$ | 0.327 | $r$ | 0.560 | $r^2$ | 0.313 |
| | $y = 3.782 + 0.523x$ std 0.543　0.067 | | | | $y = 4.109 + 0.478x$ std 0.531　0.066 | | | | $y = 4.363 + 0.474x$ std 0.544　0.067 | | | |
| Management | $r$ | 0.476 | $r^2$ | 0.227 | $r$ | 0.602 | $r^2$ | 0.362 | $r$ | 0.569 | $r^2$ | 0.324 |
| | $y = 4.545 + 0.424x$ std 0.620　0.075 | | | | $y = 4.314 + 0.464x$ std 0.488　0.059 | | | | $y = 4.363 + 0.474x$ std 0.528　0.064 | | | |
| Site condition | $r$ | 0.429 | $r^2$ | 0.184 | $r$ | 0.441 | $r^2$ | 0.195 | $r$ | 0.529 | $r^2$ | 0.280 |
| | $y = 4.842 + 0.376x$ std 0.613　0.076 | | | | $y = 4.843 + 0.381x$ std 0.599　0.074 | | | | $y = 4.045 + 0.486x$ std 0.602　0.075 | | | |
| Procurement | $r$ | 0.368 | $r^2$ | 0.135 | $r$ | 0.385 | $r^2$ | 0.148 | $r$ | 0.389 | $r^2$ | 0.151 |
| | $y = 5.608 + 0.288x$ std 0.564　0.070 | | | | $y = 5.687 + 0.290x$ std 0.528　0.067 | | | | $y = 5.696 + 0.305x$ std 0.559　0.069 | | | |
| External factors | $r$ | 0.386 | $r^2$ | 0.149 | $r$ | 0.436 | $r^2$ | 0.190 | $r$ | 0.485 | $r^2$ | 0.235 |
| | $y = 5.138 + 0.350x$ std 0.659　0.080 | | | | $y = 5.245 + 0.348x$ std 0.565　0.069 | | | | $y = 4.943 + 0.393x$ std 0.557　0.068 | | | |

Note: *sig for all factor* $\leq$ 0.005.

Given the values of r and $r^2$, all safety factors should be recognised as an integrated package to ensure the effectiveness of SS in reducing waste. The lack of commitment to any safety factors leads to disruption in the entire SS. Additionally, the values of r and $r^2$ show the presence of other factors not related to safety factors, which affect CW reduction in CPs. This result is logical.

## 5. Conclusions and Recommendations

This study identified and ranked 24 safety factors and their positive effects on reducing waste (materials, time and cost) in CPs during IPh. The conclusions drawn are as follows:

Hypotheses testing showed a statistically remarkable relationship between commitment to SS during IPh and minimising CW.

The seven most important factors that should be considered when minimising material, time and cost wastage are handling, management, factors, workers, procurement, site condition and appropriate scaffolding for SS. The lowest factors are the monitoring system, accident reports and competency of workers and ongoing training.

A model was constructed on the basis of the statistical test results and the relatively important factors. This model shows the relationship between degrees of commitment to SS and minimising waste (materials, time and cost) in CPs during IPh; it can predict the minimisation of waste in CPs during IPh by using SS.

SS should be used during IPh to minimise waste (materials, time and cost) on the basis of the developed model.

Safety training programmes must be developed for supervisors, contractors and workers to increase their skills and knowledge related to the concept and requirements of SS and CWM.

General and special conditions about safety requirements (plan, materials, equipment, implementation methods and schedule) must be confirmed in any CP.

Classification of contractors about the requirement for SS must be established in any CP.

An OSH information system must be developed. This system includes the results of the visits of OSH inspectors and the legal actions taken against the violating construction companies. Work injury data in the construction sector, compensation for these injuries and their medical examinations, occupational diseases and their causes, accident and risk records, working hours, wages and working conditions affecting OSH, the nature of work for each worker and the working register are also included.

Suggested directions for future research include the following:

1. Investigating the relationship between using SS in the design phase and minimising waste (materials, time and cost)
2. Investigating the relationship between using SS in the maintenance phase and minimising waste (materials, time and cost)
3. Developing a computerised programme to help project stakeholders calculate the relationship between the variables (safety and waste)

**Author Contributions:** K.M. designed the research, collected, and analyzed the data, and drafted the paper under supervision of A.L. and K.A.H. B.A.T. contributed in reviewing the final work to enhance/improve the outcomes.

**Funding:** The authors declare no external funding.

**Acknowledgments:** The authors gratefully acknowledge the Department of Civil Engineering at Tunis El Manar University for providing technical support to conduct this research. The authors also acknowledge the anonymous reviewers for their comments.

**Conflicts of Interest:** The authors declare no conflict of interest.

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
