# Peer review of "Implementation Phase Safety System for Minimising Construction Project Waste"

_buildings, doi:10.3390/buildings9010025_

Reviewer 1 Report

Title:

It is necessary to indicate in the title the area of application of the investigation (for example: geographical area, country, state, etc.). If the authors imply that it has universal application, they must prove it at work. It should be noted that the study is valid only for companies classified as: "first class" according to NCC

Abstract:

Authors are asked to indicate the most important results found in this research; the results should be expressed in values or percentages to facilitate the understanding and importance of the work

Keywords:

Authors are asked to sort the keywords from the most specific to the most general

Introduction:

It is necessary to avoid defining acronyms more than once (for example PC), all the work should be reviewed.

It is considered that it is necessary to conduct a more exhaustive search of information from previous investigations (it is recommended to use SCOPUS, WoS, Compendex, etc., 10 years prior) Without the integration of prior knowledge research, this work would be isolated and therefore not it would have utility.

It is notorious the lack of reference in respect of regulations for the application of safety in construction; there are in each country those regulations that should be incorporated in this investigation, for example:

https://www.osha.gov/Publications/OSHA3252/3252.html

https://www.osha.gov/pls/oshaweb/owastand.display_standard_group?p_part_number=1926&p_toc_level=1

http://dof.gob.mx/normasOficiales/4376/stps/stps.htm

Figure 2: improve the format avoiding letters to overlap the lines (check all)

Calibration of questionnaire: The authors have performed a questionnaire calibration by specialist review (it is correct); However, the authors are requested to review the following work (it has application in teaching subjects, however the procedure is valid for any questionnaire). Satisfying this requirement will allow to give more solidity and irrefutable value to the data extracted from the analysis of the professionals who carried out the survey.

https://journals.sagepub.com/doi/pdf/10.1177/2158244013499159

Include the bibliographic reference of the list of CNN companies

Equation 1, 2 and 3: It is necessary to indicate the bibliographic reference of the origin of these equations. If the equations are original to this investigation, they must be demonstrated. (check the entire document)

Figure 3. Check the horizontal axis title

3.4 Measurement and analysis of data:

It is necessary to include references that allow to validate all the statistical tests performed as correct. It is not necessary to explain them, but it is necessary to compare them with other similar investigations.

There are two Figures 3. (Check the order of all figures and tables)

It is considered that Figure 4 and 5 does not provide relevant information, it is requested to eliminate it and improve the description of the samples in the text.

Figure 6 include the value of R2 of each regression equation.

Comment made with respect to the authors. My professor of statistics (when I studied), said that statistics is the third way of telling lies (for evil, for not doing harm and the third way is by statistics.) With the above, I do not question the correct work done , and the value of the deductions that are extracted, however, for this work to have a proven value, I ask the authors to carry out field experimentation with real cases, in this way they can validate the conclusions drawn statistically and the work will have a scientific value for the construction sector.

It is requested to include a new section of real practical validation of the hypotheses taken statistically.

Conclusions:

When the authors make the requested modifications, it is requested to review and improve the conclusions again.

Bibliography:

It is considered necessary to improve or increase the number of references.

Author Response

Thank you for your very useful comments, suggestions and modification of the manuscript

The followings are the specific responses to the comments from reviewers.

Reviewer 2 Report

Authors deal with an interesting topic of construction project management by researching the safety system for minimizing construction projects waste (in material, time, and/or cost). Although the topic is very interesting there are several major insufficiencies that need to be improved. These insufficiencies can be summed up in poor scientific writing and poor paper structure.

Suggestions for improvement:

·         Check and improve English language and grammar throughout the paper

·         Paper should be set according to Journal’s template and instruction to authors (text, figures, tables, references, etc.)

·         Please reconstruct and rewrite the abstract; highlight the findings

·         Avoid “hypothesis tests” as a keyword

·         Please clearly state the research goals and hypotheses; lines 50-57 are very vague and misleading, especially related to the discussion and conclusions (see lines 311-317)

·         In literature review reflect upon other researches; avoid just numbering references such as in lines 84, 100, 106, and 109

·         Research methodology lacks consistency. This is most evident in Fig. 2. It is poorly done, unclear, and lacks several information

·         From the beginning of part Findings and discussion onward the figure numbering is wrong

·         During discussion of results, authors should reflect upon low coefficients of determination (line 453 and Table 6)

·         In conclusion use sentences rather than bullet points

Overall, article has serious flaws especially in scientific writing and needs to be rewritten according to given directions.

Author Response

(The authors gave the same response as above.)

Reviewer 3 Report

The work presents and describes various sources of construction waste. The triangulation method was used to analyze them. One of the three elements is the conducted questionnaires.

It would be interesting to present what the sample questionnaire used in the study looked like.

The results are presented clearly and comprehensively analyzed.

Author Response

(The authors gave the same response as above.)

Reviewer 4 Report

This paper investigates the relationship between the implementation of safety management systems and construction wastes. The idea that those two would be inter-related appear novel but lacking of supports. Consequently, it is very unclear why the authors hypothesised the relationship between the two in the first place, which is a significant issue for the entire work. A few more important significant comments from this reviewer are as follows: 

- The paper needs to provide much more clear evidence that there would be relationships between safety management systems and construction wastes from the beginning of the paper (e.g., research motivation). Safety system components usually would not be designed specifically for addressing the construction waste issue. 

- Another very important weakness of this work is that it does not provide clear definition of the key terms such as safety systems. The authors should provide a clear definition of the key terms, otherwise, it is very difficult to interpret the findings. 

- The information about how the data was actually collected is significantly insufficient. In the current form of the paper, it is impossible to evaluate the validity of the approaches used to collect and analyse the data. 

- Similarly, the data analysis flow and the methods need to be explained better in connection with research objectives. 

With these several important comments said, this reviewer believes that this manuscript would require a major revision before it can be re-considered for publication in the Journal, in this reviewer's opinion. 

Author Response

Thank you for your very useful comments, suggestions and modification of the manuscript

The followings are the specific responses to the comments from reviewers.

Round  2

Reviewer 1 Report

The authors have made the modifications or presented arguments. I believe that not all have been 100% adequate, but I respect the opinion and comments of the authors.

Therefore the work can be published.

Congratulations

Author Response

Thank you for your very useful comments, suggestions and modification of the manuscript.

Reviewer 2 Report

There are some technical elements that need correction, such as:

Figure 5 is missing

Figure 6: parts of this figure overlap

correct R2 values in Figure 6

additional clarification of adding references 60-69 is needed

Author Response

Thank you for your very useful comments, suggestions and modification of the manuscript

Reviewer 4 Report

This reviewer reviewed the revised manuscript once again carefully and thoroughly, and unfortunately, he found that most of his review comments from the previous round have been addressed only partially. In particular, the reason why the safety management system and construction wastes are related is poorly explained. 

Figure 1 added in this revision helps one to understand what kind of factors affect construction wastes; there is no factor directly related to safety management systems. Therefore, it is still very unclear why the authors hypothesised the relationship between SMS and CW (i.e., The three hypotheses in lines 313 - 320). This is a major issue, and it must be well addressed, and otherwise no one can clearly understand the motivation or the contribution of this work, in this reviewer's opinion. 

This reviewer also asked about how the survey data was collected, i.e., what exactly were the variables measured in the survey questionnaire, how was the survey questionnaire designed, how each variable was measured through surveys. Currently, this reviewer could not find any information about the survey questionnaire design or survey data collection process. This is another major issue in this work. 

In addition, the manuscript needs to explain better the concept of "construction wastes in terms of material, time and cost." It does not sound like a conventional definition of construction waste, and therefore a clearer explanation regarding the definition of CW is strongly required for one to be able to understand the findings and the implications.  

Author Response

Thank you for your very useful comments, suggestions and modification of the manuscript.

The manuscript has been extensively revised to improve the overall quality and at same time to take into consideration the comments and criticisms from the reviewers.
